# Genetic Analysis of Avian Coronavirus Infectious Bronchitis Virus in Yellow Chickens in Southern China over the Past Decade: Revealing the Changes of Genetic Diversity, Dominant Genotypes, and Selection Pressure

**DOI:** 10.3390/v11100898

**Published:** 2019-09-26

**Authors:** Wensheng Fan, Ning Tang, Zhihua Dong, Jiming Chen, Wen Zhang, Changrun Zhao, Yining He, Meng Li, Cuilan Wu, Tianchao Wei, Teng Huang, Meilan Mo, Ping Wei

**Affiliations:** College of Animal Science and Technology, Guangxi University, Nanning 530004, China; 2003424318@163.com (W.F.); tangningji77@163.com (N.T.); dongzhihua0818@163.com (Z.D.); chenjiming8@126.com (J.C.); wszw0719@163.com (W.Z.); zcrun628@163.com (C.Z.); heyiningf@foxmail.com (Y.H.); mengli4836@163.com (M.L.); cuilanwu@163.com (C.W.); tcwei88@126.com (T.W.); tenhhwang@163.com (T.H.)

**Keywords:** infectious bronchitis virus, genotype, entropy, molecular evolutionary rate, recombination, selection pressure

## Abstract

The high mutation rates of infectious bronchitis virus (IBV) pose economic threats to the poultry industry. In order to track the genetic evolutionary of IBV isolates circulating in yellow chickens, we continued to conduct the genetic analyses of the structural genes S1, E, M, and N from 64 IBV isolates in southern China during 2009–2017. The results showed that the dominant genotypes based on the four genes had changed when compared with those during 1985–2008. Based on the S1 gene phylogenetic tree, LX4-type (GI-19) was the most dominant genotype, which was different from that during 1985–2008. The second most dominant genotype was LDT3-A-type, but this genotype disappeared after 2012. New-type 1 (GVI-1) isolates showed increasing tendency and there were four aa (QKEP) located in the hypervariable region (HVR) III and one aa (S) insertion in all the New-type 1 isolates. Both the analyses of amino acid entropy and molecular evolutionary rate revealed that the variations from large to small were S1, E, M, and N. Purifying selection was detected in the S1, E, M, and N gene proteins, which was different from the positive selection during 1985–2008. Six isolates were confirmed to be recombinants, possibly generated from a vaccine virus of the 4/91-type or LDT3-A-type and a circulating virus. The estimated times for the most recent common ancestors based on the S1, E, M, and N genes were the years of 1744, 1893, 1940, and 1945, respectively. Bayesian skyline analysis revealed a sharp decrease in genetic diversity of all the four structural genes after 2010 and since late 2015, the viral population rapidly rose. In conclusion, the IBVs circulating in southern China over the past decade have experienced a remarkable change in genetic diversity, dominant genotypes, and selection pressure, indicating the importance of permanent monitoring of circulating strains and the urgency for developing new vaccines to counteract the emerging LX4-type and New-type IBVs.

## 1. Introduction

Infectious bronchitis (IB) is one of the major viral diseases affecting the poultry industry globally. The causative agent avian infectious bronchitis virus (IBV) is a member of the genus *Gammacoronaviruses*, subfamily *Coronavirinae*, family *Coronaviridae* and is prone to mutate. There are multiple genotypes and serotypes of IBV isolates identified worldwide and limited cross-protection confers between serotypes of IBVs [1,2,3,4,5,6], which poses great challenge to the control of IB by vaccination.

The IBV has a single-stranded RNA genome of approximately 27.6 kb in length [7] and encodes four structural proteins: the spike (S), envelope (E), membrane (M), and nucleocapsid (N) proteins [8,9]. The S protein is cleaved into subunits S1 and S2 by proteases [10,11]. The function of the four structural proteins has been extensively reviewed by others [12,13,14]. The genetic analysis based on the S1 gene has become the primary method of classifying IBV strains because of variability and functional importance [15]. However, those previous IBV molecular characterizations that were merely focused on the analysis of the S1 gene or partial S1 gene sequence could not explain the changes in the serotypes and pathotypes of IBV variants [16]. Sometimes, single gene analysis even is misleading. Therefore, it is necessary to analyze all the structural protein-coding genes S1, E, M, and N simultaneously in order to obtain comprehensive genetic information and molecular mechanism of variation of circulating IBV isolates.

A variety of IBV genotypes and variants are distributed globally. So far, a total of seven genotypes comprising 35 distinct viral lineages have been defined worldwide based on the complete S1 gene sequences [17]. IBV strains within a certain country or region are unique even though many countries share some common antigenic types. For example, two distinct lineages that fall in two different genotypes—GI-21 and GII-1—were identified as unique to Europe. GI-9, GI-27, and GIV-1-genotypes have been implicated in widespread disease disseminations and persistent virus infections in north American [16,18,19]. GI-1, GI-11, and GI-16 are currently circulating in South American flocks [20]. LX4-type (QX-type or GI-19) which was firstly isolated in China in 1996 spread westward invading Russia, Middle East, and Europe, and then becoming the most prevalent genotype in many countries, such as Korea, Russia, Iran, Italy, UK, Malaysia, Sudan, and so on [7,8,21,22,23,24,25]. Nowadays, LX4-type (GI-19) and CK/CH/LSC/99I-type (GI-22) appear to be the dominant viruses based on the S1 gene in China [18]. Because of differences in breeding variabilities and feeding patterns of chickens in China, the characteristics of circulating IBVs at different times and in different regions are variable. Hence, it is necessary to conduct long-term tracking of the IBV circulating isolates in specific geographic regions or countries for the effective control of IB [4,18,26].

The appearance of IBV variants was related to high mutation and recombination rates, which result in the generation of genetic diversity and phenotypic heterogeneity. However, IBV evolution is not driven by genetic drift alone. Evolution involves two fundamental steps—i.e., generation of genetic diversity and selection [27]. The selection process was affected by multiple factors, such as immune responses, the microenvironment of infected hosts, physical and biosafety conditions [16,27]. Vaccines not only give rise to new variants through recombination, but also impose selection pressure on the evolution of field strains [28,29,30]. It is essential for appropriately controlling and prevention of the disease to understand the evolution of IBV [27].

Southern China is the major production region of Yellow chickens (4.0 billion in 2018 and made a proportion of 37% of the total chickens) in the country. Guangxi province, located in southwest of China, has the biggest production of local breeds of chickens (0.75 billion birds in 2018) [31], and most of the birds are free-range and raised for a longer time (about 120-day-old) at a rather high density, and the chicks are raised in relatively closed environment and lack of ventilation during brooding [32,33]. Also, flocks of various companies, with different chicken breeds and differing ages and vaccination programs located in the same areas are common [32]. These situations surely increase the odds of multiple-infection of several IBV strains including the vaccine strains or/and field strains in flocks of birds [34]. Despite the widespread use of Mass-type (Massachusetts genotype) (H120, H52, Ma5, M41, and W93), 4/91, LTD3-A live vaccines and inactivated vaccines, IB has been a continuing problem in vaccinated flocks in these regions [35]. Our previous studies found the serotype and genotype diversity of Guangxi IBVs from 1985 to 2008 and the important role of vaccine strains in the emerging of new IBV strains via recombination [1,2,4]. However, the comprehensive genetic information of circulating IBV strains in this region was unavailable over the past decade. Hence, we continued to carry out the genetic analysis of 64 IBV isolates during 2009–2017. The aim was to track the genetic evolutionary trends of IBV field strains circulating over the past decade and their possible causes through relatively new and comprehensive analyses of virus genes, and then provide valuable reference and countermeasure against IBV field breakouts in southern China.

## 2. Materials and Methods

### 2.1. Virus Isolation and Propagation

Sixty-four IBV strains isolated from flocks of Yellow chickens during 2009–2017 were analyzed in the present study (Table 1). All IBV field isolates were obtained from the birds of previously vaccinated flocks with H120, LDT3-A and/or 4/91 vaccines and experienced clinical signs of the IBV infection. All IBVs were isolated and propagated as previously described [4].

### 2.2. Primers for S1, M, N, and E Genes Amplification

The entire S1, E, M, and N genes were amplified for each IBV strain and the primers used were designed as previously described [4,36]. The anticipated amplification segments for the S1, E, M, and N genes are 1760 bp, 633 bp, 750 bp, and 1300 bp in lengths, respectively.

### 2.3. RNA Extraction and Amplification of S1, E, M, and N Genes

Viral RNA was extracted and the first cDNA strand was synthesized as previously described [4]. The PCR conditions for the S1, M, and N gene amplification were the same as previously described [4]. The PCR conditions for the E gene amplification were 95 °C for 5 min, 35 cycles of 95 °C for 30 s, 52 °C for 30 s, and 72 °C for 30 s, followed by 72 °C for 6 min. The PCR products were analyzed on 1.0% agarose-gel electrophoresis.

### 2.4. Gene Sequencing, Alignments, and Phylogenetic Analysis

The PCR products of S1, E, M, and N genes were sequenced by Beijing Genomics Institute (BGI) (Shenzhen, China) after cloning. The open reading frames of 64 IBVs were determined and their nucleotide sequences were submitted to GenBank database and assigned accession numbers (Table 1). Sequences of 43 reference IBV strains (with the exception of 42 strains for E gene) retrieved from the GenBank database were used (Appendix A). The 27 IBVs isolated during 1985–2008 in Guangxi [4] were also analyzed together in order to get a general profile of IBV evolution. The nucleotide and deduced amino acid (aa) sequences of the S1, E, M, and N genes obtained from the IBV isolates were aligned using the Editseq program in the Lasergene package (DNASTAR Inc., Madison, WI, USA) and compared to those of IBV reference strains representing the main well-established lineages and genotypes using the MegAlign program in the same package. To ensure the scientificity and reliability of the results, we extracted the S1, E, M, and N genes from the complete genome sequences of reference strains. Phylogenetic trees based on the aa sequences of S1, E, M, and N genes were constructed using MEGA version 6.06 according to previous description with nodal support values obtained by posterior probabilities and 1000 bootstrap replicates [37].

### 2.5. Recombination Detection

Aligned nucleotide sequences of S1, E, M, and N genes were subjected to the Recombination Detection Program (RDP 4 Version 4.95) to detect potential recombination events by seven algorithms (RDP, GENECONV, Bootscan, MaxChi, Chimera, SiScan and 3Seq) in RDP 4.95 [4]. The detection of recombination breakpoints by at least four of these methods were considered as confirmation of any putative recombination event. The potential recombination events and breakpoints were further verified by similarity plots (SimPlots) analysis in SimPlot version 3.5.1 [7,14].

### 2.6. Analysis of Entropy of Amino Acid Sequences

Entropy of amino acid sequences within the S1, E, M, and N proteins of IBV isolates was calculated by BioEdit version 7.1.11.0 in order to understand the variation degree of these four structural protein genes [4].

### 2.7. Analysis of Positive Selection

Positive selection and positively selected sites within the S1, E, M, and N proteins were analyzed by the single-likelihood ancestor counting (SLAC), fixed effects likelihood (FEL), and internal fixed effects likelihood (IFEL) methods of Datamonkey version (http://www.datamonkey.org/26/08/2019) [26] to detect whether these proteins have undergone positive selection. The recombinants were excluded in order to reduce a false detection of positive selection.

### 2.8. Prediction of N-Glycosylation Sites

The potential N-glycosylation sites were predicted within the S1, E, M, and N proteins. The analysis was performed using NetNGlyc server 1.0 software available at http://www.cbs.dtu.dk/services/NetNGlyc [38].

### 2.9. Analysis of Molecular Evolutionary Rate, the Most Recent Common Ancestor, and Population Size

The results of aligned sequences were computed by multiple alignment with fast Fourier transformation (MAFFT) [39]. The nucleotide substitution process was modelled independently for each partition with GTR (general time reversible) + G (gamma distribution with a discrete) + I (proportion of invariant sites) based on AIC by the jModel Test 2.1.7. Bayesian tree reconstructions were performed in BEAST version 1.8.2. A Bayesian skyline coalescent model and strict molecular clock were selected. The Bayesian Markov chain Monte Carlo (MCMC) chains of the S1, E, M, and N genes were run for 300 million, 20 million, 100 million, and 200 million generations, respectively. Results were analyzed using Tracer version 1.5 and confirmed convergence of MCMC chains with 10% of each chain discarded as burn-in and sampled every 10,000 steps [30]. Statistical uncertainty (reflected calculating 95% high probability density (HPD) values) and convergence (reflected calculating effective sample size) in parameter estimates were evaluated in Tracer version 1.5 program. The posterior sets of trees were summarized as a maximum clade credibility (MCC) tree using Tree Annotator version 1.8.2 with 10% burn-in and then displayed the created MCC tree using FigTree version l.4.3 [40]. The mutation rates and the most recent common ancestor (tMRCA) of the aligned sequences were estimated. The change of effective population size over time was inferred by Bayesian skyline plots (BSP).

## 3. Results

### 3.1. Alignment Analysis of S1, E, M, and N Gene Sequences

The nucleotide and deduced aa sequence similarities of the S1, E, M, and N genes among the 64 isolates during 2009–2017 were 63.2–100% and 57.2–100%, 80.1–100% and 78.3–100%, 85.7–100% and 86.7–100%, and 84.6–100% and 88.3–100%, respectively. Compared with H120, the isolates GX-QZ150024, GX-LZ160322, and GX-NN171125 have higher amino acid sequence similarities of 97.4–99.8%, 93.6–100%, 99.1–100%, and 94.4–100% in the S1, E, M, and N genes, respectively. Within the S1 gene, there were 11 different nucleotide lengths (from 1590 to 1638 bp), and the most common lengths were 1620 bp (35/64 isolates, 54.69%) and 1626 bp (18/64 isolates, 28.13%). There were eight types of S protein cleavage site motifs found: RRFRR (22/64), HRRRR (18/64), RRLRR (9/64), RRSRR (7/64), HRRKR (5/64), HRIKR (1/64), HRSKR (1/64), and RKRKR (1/64) among the isolates. There were four aa (QKEP) (located in the hypervariable region (HVR) III) and one aa (S) insertion found in the S1 genes of six isolates (GX-NN09032, GX-NN120079, GX-NN120084, GX-NN120089, GX-QZ130064, and GX-QZ130065), respectively (Appendix A).

### 3.2. Phylogenetic Analysis

The S1 gene phylogenetic tree showed that the IBV isolates during 2009–2017 were divided into eight distinct groups (Figure 1a). Counts of 17, 14, 7, 7, and 6 out of 64 IBV isolates during 2009–2017 belonged to LX4-type (QX or GI-19), LDT3-A-type (GI-28), 4/91-type (GI-13), Mass-type (GI-1), and CK/CH/LSC/99I-type (GI-22), respectively. Five isolates (GX-NN1014, GX-NN130059, GX-NN170502, GX-NN170829, and GX-NN171123) in recent years were grouped with Taiwan reference strains TW2575/98 as Taiwan-I-type (GI-7). Six isolates (GX-NN09032, GX-NN120079, GX-NN120084, GX-NN120089, GX-QZ130064, and GX-QZ130065) and two isolates (GX-NN130021 and GX-YL150727) showed considerable low similarities (57.2–68.1%, 58.8–67.3%) with the above genotypes and belonged to two separate groups New-type 1 (GVI-1) and New-type 2 (GVII-1).

The phylogenetic trees of E, M, and N genes of the 64 isolates were segregated into six, four, and six unique groups, respectively (Figure 1b–d). And their phylogenetic trees exhibited considerably different topology compared with that of the S1 gene. No obvious geographic differences were found among the 64 isolates, while there was a high degree of sequence identity among the isolates in the same period of time (Appendix A).

The percentages of different genotypes based on S1, E, M, and N genes of IBV isolates in different years were summarized in Figure 2. Based on the S1 gene, the CK/CH/LSC/99I-type was the predominant genotype during 1985–2008, but the LX4-type was the predominant genotype circulating in the field during 2009–2017. The LDT3-A-type was the second most dominant genotype, but this genotype disappeared after 2012. Based on the E and M genes, the CK/CH/LSC/99I-type was the predominant genotype during 1985–2008, while the CK/CH/LSC/99I-type and LX4-type were the predominant genotypes during 2009–2017. Based on the N gene, the CK/CH/LSC/99I-type and the LX4-type were the predominant genotypes during 1985–2008, but only the LX4-type was the predominant genotype during 2009–2017. Therefore, our results demonstrated that the CK/CH/LSC/99I-type isolates were the predominant IBVs according to the phylogenetic study of the S1, E, M, and N genes from 1985 to 2008. Thereafter, the proportion of LX4-type, LDT3-A-type, and New-type strains increased over time. The LDT3-A-type isolates were the predominant IBVs between 2009 and 2010. The LX4-type isolates had become the predominant IBVs since 2011.

### 3.3. Analysis of Recombinants

Recombination events of the S1, E, M, and N genes of 64 isolates were examined using the RDP in the study. The results showed that recombinant events were found in the S1 gene of six isolates (Figure 3 and Table 2). GX-NN1011 was derived from recombination between the two LX4-type strains GX-NN6 (major parent) and GX-NN-9 (minor parent). GX-NN1201 was derived from recombination between LX4-type strain QX (major parent) and CK/CH/LSC/99I-type strain SAIBK (minor parent). GX-NN130048 was derived from recombination between LX4-type strain GX-HC1006 (major parent) and 4/91 strain (minor parent). GX-NN120091 was found to be a recombinant between the New-type 1 isolate GX-NN120089 (major parent) and the vaccine strain LDT3-A (minor parent). GX-NN130003 was found to be a recombinant isolate formed by a major parent isolate GX-YL11072 (LDT3-A-type) and a minor parent isolate GX-YL170717 (CK/CH/LSC/99I-type). The potential parents of GX-YL170805 were proved to be a major parent isolate GX-NN1014 (TW2575/98-type) and a minor parent isolate GX-NN10 (4/91-type). In addition, it was found that A–T rich hotspot sequence ATTTT (T/A) was near the breakpoint site of the S1 subunit gene in all of recombinant isolates except GX-YL170805 (Appendix A). In order to confirm the results of the RDP analysis, genomic sequences analyses of the six IBV isolates were carried out by the Simplot software, and the results were consistent with those of the RDP analysis (Figure 4). Therefore, the dominant recombinants were the LX4-type isolates from 2009 to 2017.

### 3.4. Analysis of Entropy of Amino Acid Sequences

The entropy of the amino acid sequences of S1, E, M, and N genes showed that there were many high entropy amino acid sites within the S1 gene, but a few high entropy amino acid sites were scattered within the E, M, and N genes (Appendix A). The average entropy values sorted from large to small as follows: S1 (0.4047), E (0.2453), M (0.1149), and N (0.1094). The percentages of entropy value which was bigger than 0.4 in S1, E, M, and N genes were 37.23% (207/556), 24.17% (29/120), 10.57% (24/227), and 11.25% (46/410), respectively. Therefore, the largest variation was observed in the S1 gene, and the E gene was also more variable than the M and N genes.

### 3.5. Positive Selection of the S1, E, M, and N Proteins

The selection profile of S1, E, M, and N proteins of totally 80 (with 11 deleted recombinant strains) IBVs were showed in Table 3. The dN/dS ratio of S1, E, M, and N proteins of eighty isolates were 0.283, 0.266, 0.146, and 0.147, respectively, indicating that the S1, E, M, and N proteins of these IBV isolates had evolved under purifying selection (Table 3). However, a few positively selected sites were detected although most sites were under neutral selection and purifying selection (Table 3 and Appendix A). Aa residues 19, 39, 106, 119, 155, and 503 of the S1 protein were consistently highlighted by positive selection models (SLAC, FEL, and IFEL) as positive selection sites. Similarly, residue 119 of the E protein, residues 12, 16, 17, 47, and 151 of the M protein, residues 7, 10, 235, 342, and 345 of the N protein were identified as positively selected sites. In addition, all of the positively selected sites within S1, E, M, and N genes had high entropy values of larger than 0.6 (Appendix A).

### 3.6. Prediction of N-Linked Glycosylation Sites in S1, E, M, and N Proteins of IBVs by NetNGlyc Server

The results showed that there were 14–22, 1–3, 1–2, 1–3 potential glycosylation sites within the S1, E, M, and N proteins (except that GX-NN120084 strain did not have a glycosylation site in the N protein). The comparison of the estimated *N*-glycosylation sites of S1 protein between the isolates and the reference strains showed that all the strains presented one *N*-glycosylation site at the residue 239/240 (NFSD) (except CK/CH/LSC/99I-type (GI-22) strains). And at the residue 427/428 (NITL), all the strains presented one N-glycosylation site (except Taiwan-type (GI-7) and New-type 1 (GVI-1)). Similarly, all the strains presented one N-glycosylation site at E protein residue 11/12 (NGSF) and residue 5/6 (NKTL) (except CK/CH/LSC/99I-type and Conn-type), all the strains presented one N-glycosylation site at M protein residue 3/4/6 (NCTL) and at N protein residue 32 (NASW) (except GX-NN120084 strain) (Table 4 and Appendix A).

### 3.7. Molecular Evolutionary Rate, the Most Recent Common Ancestor, and Population Size Analysis

The mean substitution rates for the S1, E, M, and N genes of the epidemic isolates during 1985–2017 were calculated to be 4.6 × 10^−3^, 4.3 × 10^−3^, 3.9 × 10^−3^, and 3.7 × 10^−3^ substitutions/site/year (s/s/y), respectively (Table 5), indicating that S1 gene is the most easily mutated and N gene is the most stable among the four structural genes. The estimated times for most recent common ancestor (tMRCA) of S1, E, M, and N genes were before 1744.97, 1896.7, 1940.3, and 1945.92, respectively (Figure 5). Based on S1, genotypes of CK/CH/LSC/99I (GI-22), New 1 (GVI-1), Mass (GI-1), LX4 (GI-19), 4/91 (GI-13), Taiwan (GI-7), LDT3-A (GI-28), and New 2 (GVII-1) were dated back to 1914.5, 1954.33, 1971.15, 1971.82, 1983.26, 1985.05, 1991.66, and 2004.08, respectively. Based on E, genotypes of CK/CH/LSC/99I, 4/91, Mass, Conn, LX4, LDT3-A were dated back to 1937.28, 1984.79, 1994.89, 1997.21, 2008.65, and 2013.62, respectively. Based on M, genotypes of QX, CK/CH/LSC/99I, Mass, 4/91 were dated back to 1959.94, 1965.68, 1978.52, and 2003.38, respectively. Based on N, genotypes of LX4, New, CK/CH/LSC/99I, Peafowl/GD/KQ, Mass, and 4/91 were dated back to 1980.55, 1983.91, 1984.33, 1992.82, 2002.92, and 2006.4, respectively.

The reconstruction of population history was assessed using a Bayesian skyline plot coalescent model. Results showed that the effective population size of S1, E, M, and N genes of IBVs was featured by a continuous and slow reduction in viral population size between 1970s and 2010s. A more obvious decrease was observed between 1990s and 2000s base on the S1 gene (Figure 6a). Remarkably, a sudden and sharp decline in the effective population size was observed after 2010s base on the S1, E, M, and N genes. However, since late 2015, the viral population rapidly rose (Figure 6).

## 4. Discussion

Southern China is the largest region of yellow chickens produced in China. Despite extensive vaccination, IB continues to be a serious problem. In the present study, 64 strains of IBV were isolated from diseased chicken flocks in southern China during 2009–2017 and the genetic properties of the entire S1, E, M, and N genes analyzed. To increase our insight into the comprehensive epidemiological situation and evolutionary trend of IBV in southern China, 27 IBV isolates from 1985 to 2008 in southern China were also used to analyze together. The 43 representative reference strains included the live vaccine strains commonly used and the prevalent strains in China and other countries worldwide, which represented the main well-established lineages and genotypes in recent years in China [17]. Our results indicated that there was a remarkable change in genetic diversity, dominant genotypes, and selection pressure of IBV strains in southern China over the past decade compared with the previous period of 1985–2007.

The co-circulation of multiple IBV genotypes and the increasing of IBV variants have resulted in great challenges for the controlling IB through vaccination. In our study, multiple IBV genotypes were also identified in Guangxi over the past decade. A total of eight, six, four, and six genotypes were identified based on the S1, E, M, and N genes, respectively. Therefore, there is still ongoing genetic diversity of IBVs in southern China, which is the same as previous epidemics in this region and other parts of China [4,15,35]. Interestingly, the predominant genotype changed from CK/CH/LSC/99I-type during 1985-2008 to LX4-type and LDT3-A-type during 2009–2017 based on the S1 gene; the predominant genotype changed from CK/CH/LSC/99I-type during 1985–2008 to LX4-type during 2009–2017 based on the E, M, and N genes, indicating there were remarkable changes in the genotypes of prevalent IBV isolates in southern China during 2009–2017 compared with the period of 1985–2008. Thus, new vaccines suitable for effective control the local predominant IBV isolates should be chosen/or developed in southern China.

LX4-type was the most dominant genotype in our study, which agreed with many previous descriptions [15,30,35,41]. Surprisingly, the second most dominant genotype of IBV circulating in southern China was LDT3-A-type, which was a pandemic type and frequently isolated from chicken flocks in China [15,42,43]. LDT3-A commercial live vaccine has been issued by the official authority in China since 2011 [6]. The re-isolation of vaccine strain is possible when LDT3-A live vaccine strain is extensively used. We noticed that among the 14 LDT3-A type isolates during 2009-2017, four, eight and two strains were isolated in 2010, 2011, and 2012 respectively. However, only one LDT3-A type strain was isolated during 1985–2008. We are not sure whether or not the 10 strains of LDT3-A type isolated between 2011 and 2012 are re-isolation of vaccine strains, but it seems that the LDT3-A type vaccine could provide sufficient protection against the circulating homologous field strains in southern China because none of LDT3-A genotype strains were isolated after 2012. At the same time, we found the E genes of most LDT3-A type isolates belonged to the CK/CH/LSC/99I-type, and the M and N genes of most LDT3-A type isolates belonged to LX4 (QX)-type. These results implied that LDT3-A-type might isolate the recombinants from LDT3-A vaccine strain and other circulating IBV strains. Considering that the LDT3-A types were occasionally isolated in other regions of China recently [43], the LDT3-A type isolates in southern China still need to be further monitored.

The Taiwan-type IBVs were firstly isolated in Taiwan in 1990s and have been divided into the Taiwan-I and Taiwan-II subgroups [44,45]. More studies reported that an increasing number of Taiwan-type strains have been isolated in southern China in recent years [15,35,46,47]. Taiwan-II-type strains (GX-XD and GX-G) were isolated in 1988 and no Taiwan-II-type strain was isolated in Guangxi after that [4], but five Taiwan-I-type strains were isolated during 2010–2017. In our study, 12.5% (1/8), 9.1% (1/11), and 20% (3/15) of Taiwan-I-type strains were isolated in 2010, 2013, and 2017, respectively. It seemed that the Taiwan-I-type strains were increasing in recent years. Evidences proved that the Taiwan-I-type strains were widely spread from Taiwan to Guangdong, Guangxi, Fujian, Sichuan Hunan, Zhejiang, Yunnan province and so on [15,46,48,49,50], and the increasing isolates demonstrated that the currently used vaccines were lacking protective efficacy against the Taiwan-I-type strains. The recombinants between Taiwan-I-type and LX4 (QX) causing severe economic losses have been reported [51]. How the Taiwan-type strains spread from Taiwan to mainland China remains unknown. Recently, wild birds have been of concern as natural IBV carriers, since infected birds have been showed to carry the viruses over long distances [8,23,52,53]. Therefore, monitoring of wild birds should be needed in further. Of course, IBV live vaccines and poultry products will also need to be followed.

The earliest strain of New-type 1 (GVI-1) was isolated in 2007 [54] and has in recent years spread to many chicken flocks [5,15,30,43,55]. Surprisingly, majorities of the New-type 1 strains from other studies were isolated from south China [2,5,15,55]. Recently, another novel genotype VII (GVII-1) was reported [17]. Both these two New-type viruses were identified in our study and the total number of their strains showed New-type was the third most dominant genotype. We found that there was a four aa (QKEP) insertion located in the HVR III and one aa (S) insertion in all the New-type 1 isolates. Interestingly, the four aa (QKEP) were predicted B cell epitopes by the BepiPred /IEDB tool and half of the New-type viruses’ serotype were different from that of 4/91 or other vaccine strains [56]. The aa mutation in the HVR of the S1 gene maybe lead to the occurrence of new serotype and immunity escape. The origin of New-type 1 viruses in China remains unknown. Interestingly, the “novel” isolates were shown to undergo recombination in our study. The New-type 1 isolate GX-NN120089 was a major parent to the recombinant LDT3-A-type isolate GX-NN120091. To the best of our knowledge, this first report of a recombination event needs to be further investigated. Analyzing the complete-genome and identifying the antigenicity and pathogenicity of New-type 1 isolates will be further studied. It is uncertain whether the New-type 1 would become the predominant type in China in further, but it continues circulating in the field suggests that the New-type 1 IBVs are still endemic in China and more attention should be paid to them.

High rates of recombination result in IBV variability. Many recombinants were confirmed. Some recombination occurred between field and vaccine viruses [5,14,15,26], some occurred between field isolates [26,35,43]. Six recombinants were identified in this study and three had a major parent of the LX4-type, and two with minor parents of 4/91-type genotype. Our previous studies confirmed five recombinants, which were between the vaccine strain 4/91 and the CK/CH/LSC/99I-type field strain GX-YL2 during 2006–2007 [4]. Therefore, we have identified 11 recombinants so far and 7 recombinants were derived from the 4/91-type strain during 1985–2017. High frequencies of recombination between vaccine and field strains have been reported frequently worldwide [4,5,15,22,57,58]. The 4/91 vaccine strain has been commonly used in China for a long time, so it is not surprising that 4/91-type recombinants were found in the field. However, there were more recombinants involved in the 4/91-type vaccine strain when compared to the Mass-type vaccine strains’ recombinants according to our and other previous reports [4,14,15,18,59], although both Mass-type and 4/91-type vaccine strains were widely used in China. The phenomenon may have the following three explanations. Firstly, incomplete protection of 4/91-type vaccine strain against heterologous strains might result in co-infection of vaccine and heterologous strains in the same birds and eventually led to recombination. Secondly, the 4/91-type vaccine strain recombinants may be more likely to escape vaccine immunization than Mass-type vaccine strain recombinants. Thirdly, the 4/91-type vaccine strain persisted longer in the immunized birds than the Mass-type vaccine strain, increasing the likelihood of co-infection with the latter infected strain/strains, as a previous report showed that 4/91 vaccine could persist in birds for 40 days [60]. The complete genome, pathogenicity, and immunogenicity of 4/91-derived recombinants should be assessed in further studies. The isolation ratio of 4/91 (GI-13) genotype strains was relatively stable recently. Therefore, in order to reduce the recombination occurred between field and vaccine viruses, the multi-valent live vaccine combined with 4/91 strain or other new vaccines (such as LDT3-A strain, LX4 strain, and so on) should be used with caution and thoughtful consideration.

In order to understand the variation degree of the S1, E, M, and N genes and the correlation between gene variation and selection pressure, the analyses of entropy of amino acid sequences, molecular evolutionary rate, and positive selection were carried out in our study. The results showed that mean substitution rates were between 3.7 × 10^−3^ and 4.6 × 10^−3^ substitutions per site per year in different genes, which is comparable with previous description [58]. Both the average entropy and mean substitution rates sorted from large to small as follows: S1, E, M, and N genes, and the dN/dS ratios of S1, E, M, and N genes were 0.283, 0.266, 0.146, and 0.147, respectively. The dN/dS ratios of S1, E, M, and N genes were less than 1, meaning that the analyzed region were under purifying selection. Therefore, our results indicated that the degree of variation sorted from large to small as follows: S1, E, M, and N genes and there was a positive correlation between selection pressure and gene variation. It is known that S1 genes had the largest variation, but the mutation of E gene has not been paid enough attention. A recent study reported that the E gene evolved at the fastest rate among four structural protein-coding genes [30]. The E gene mutation needs to be focused on in the future. To our knowledge, it is the first report to analyze systematically the variation degree of the four structural genes of IBVs and correlation between gene variation and selection pressure.

As mentioned above, the S1, E, M, and N genes of Guangxi IBVs during 2009–2017 were under purifying selection. Despite that the four structural genes underwent purifying selection, there were six, one, five, and five positively selected sites within the S1, E, M, and N genes respectively. The dN/dS value and positively selected sites of Guangxi isolates were different from those of isolates from other countries and regions [26,30,38]. In addition, all of the positively selected sites in S1, E, M, and N genes had high entropy values of bigger than 0.6. The accumulation of amino acid variation will have an important effect on the gene characteristics and evolutionary direction of viruses [4]. Therefore, more attention should be paid to those positively selected sites with high entropy values.

In the present study, positive selection was not detected in the S1, E, M, and N proteins of IBV isolates in southern China during 2009–2017 although there were high number of mutations. Zhao et al. also found that purifying selection was the main evolutionary pressure in the protein-coding regions in China over the past two decades [30]. However, positive selection was detected in the S1, M, and N proteins of IBV isolates in southern China during 1985–2008 according to our previous investigation [4]. Therefore, positive selection in the S1, M, and N proteins in different periods in southern China was changeable. This phenomenon was also seen in some other previous reports [61,62,63]. The change in selection pressures may be due to immune responses induced by multiple types of vaccines, the microenvironment of infected hosts, or physical and biosafety conditions [16].

Molecular evolutionary rate analysis can accurately estimate the molecular dating and it was used to determine successfully the timing of ancestors of classical swine fever virus (CSFV) [64], SARS virus [65], and IBV [30,58]. According our previous investigation [1], the first IBV strain was isolated in 1985, demonstrating that the IBV was already present at that time. This study further backdated the IBVs emergence of more than 30 years (during 1985–2017) and indicated that the most recent ancestor of Guangxi IBV isolates based on the genes S1, E, M, and N existed in the early 1744, 1893, 1940, and 1945, respectively. Different structural protein genes of the IBVs had different time of the most recent ancestor. This phenomenon was also reported in other viruses [51,66,67]. The data of tMRCA estimated from our study suggested IBV has been already circulating in field for a period before its first outbreak. Interestingly, the evolution rates sorted from large to small as follows: S1, E, M, and N genes; the similarity plot data showed high genetic divergence and recombination was found in the S1 gene of IBV isolates, which agreed with the dates of most recent ancestor. Although IBV was discovered in 1930s, it is thought that it may have a long history as a species. The similar results were reported in other viruses [66,67]. In our early study, maybe unavailability of advanced diagnostic and sequencing technologies, the backyard poultry farming, inefficient surveillance systems, and limited financial resources resulted in the non-detection of IBV isolates for such a long time in Guangxi.

Bayesian skyline analysis of S1, E, M, and N genes of IBVs revealed that a persistent and slow decrease in relative genetic diversity between 1970s and 2010s, mirroring the implementation of Mass- or 4/91-based vaccine was effective in controlling the virus in field conditions to some extent, although not being able to eradicate the pathogen. Analysis of S1 showed a more obvious decrease between 1990s and 2000s. The live vaccine H120 has been used in China for many years and the 4/91-like live vaccine was also commonly used in China since 1990s even without official authorization. Therefore, the 4/91-like vaccine was effective between 1990s and 2000s. Surprisingly, a sudden and sharp decrease in genetic diversity base on the S1, E, M, and N genes was observed after 2010, mirroring the implementation with a newly registered LDT3-A-based vaccine. In our study, the LDT3-A-type disappeared after 2012. LDT3-A-like strains were also rarely isolated in other recent studies [30,43]. It was reported that the change of viral population size had a strong association with the vaccine administration/withdrawal [68,69]. Therefore, it can be speculated that the LDT3-A vaccine could provide a better protection against the circulating strains in China. However, since late 2015, a sharp increase was observed in viral population size, which may suggest the implementation of current vaccines could not provide higher protection against the circulating strains. The LX4-type isolates had become the predominant IBVs since 2011 in our study. The proportion of LX4-like genotype strains increased over time in China [30]. Increasing of LX4 genotype viruses over time agreed with the rapid rise in the viral population since late 2015. Our previous study showed that the H120, 4/91, or LDT3-A vaccines could not provide complete protection against the prevalent local strains of IBV (such as LX4 genotype isolates) in southern China [6]. Therefore, it is predicted that the epidemic trend of LX4 type strain will not decline in the next few years and will become more and more intense. Therefore, new LX4 type vaccine was urgently needed. LX4 live vaccine was officially authorized in China last year, but has not yet come into the market. LX4 commercial vaccine urgently needs to go to the market in China.

Taken together, our results of genetic analyses of the four structural protein genes S1, E, M, and N clearly demonstrated that there has been a remarkable genetic change of IBV in southern China over the past decade. The coexistence of multiple genotypes of IBV, the changes of dominant genotypes in different periods, the emergence of new genotypes, the recombination of vaccine strains with the field strains, the rapid evolving rate of IBV, the ever-changing of viral population size, and the presence of multiple positively selected sites in the structural proteins suggest a complexity of IBV epidemic strains in southern China. This epidemiological complexity may be caused by multiple factors: simultaneous vaccination of multiple live vaccines, illegal use of live vaccines, large number of avian varieties, raising of chickens at a rather high densities and the patterns of raising (free-range style and lack of biosecurity), etc. The complexity also suggests the importance of molecular epidemiological investigation and permanent monitoring of circulating strains as well as more serious challenges for IB prevention and control. Based on the present results, it is predicted that LX4-type will remain to be dominant genotype in near future and New-type IBV isolates will be more and more prevalent in the future. Therefore, it is necessary to choose and/or develop new vaccines to counteract the IBVs of these two genotypes. In addition, multivalent new vaccines should be developed and rational modified vaccination strategies and the strict biosafety should be observed. To our knowledge, this is the first report showing the changes of genetic diversity, dominant genotypes, and selection pressure of IBV strains. The present study extends our knowledge about past and present IBV variability in southern China, providing valuable reference and countermeasures against IBV field breakouts in China, also emphasizing the importance of constant dynamic surveillance of the circulating isolates.

## 5. Conclusions

In conclusion, our results indicate that the IBV strains in southern China have experienced a remarkable change in genetic diversity, dominant genotypes, and selection pressure, indicating the importance of permanent monitoring of circulating strains and the urgency for developing new vaccines which counteract the emerging LX4-type and New-type IBVs and should be resistant to recombination and mutation.

## Figures and Tables

**Figure 1 viruses-11-00898-f001:**
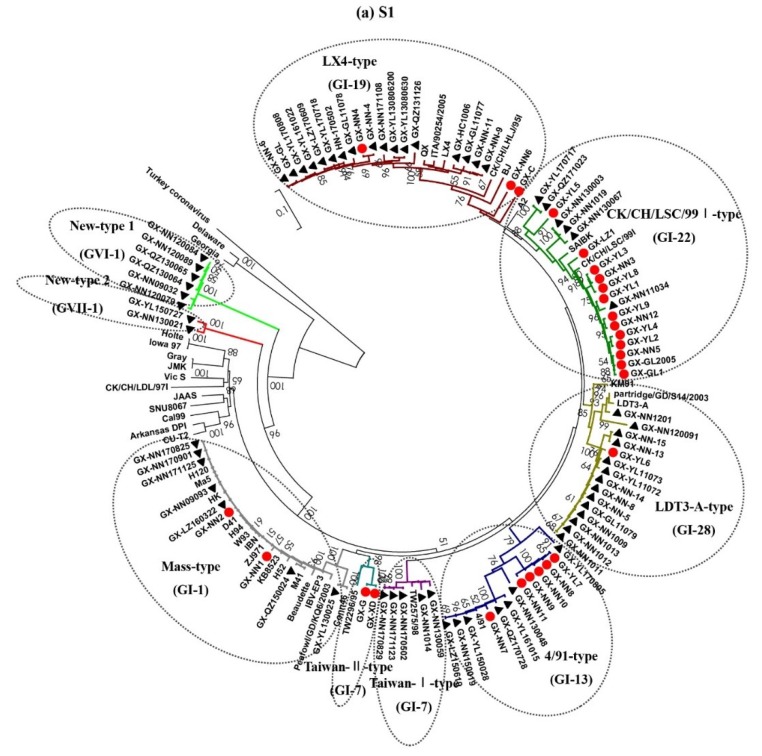
Phylogenetic trees of gene S1 (**a**), E (**b**), M (**c**), and N (**d**) of IBVs, where the 64 IBV strains isolated during 2009–2017 are marked with filled black triangles and 27 IBV strains isolated during 1985–2008 are marked with filled red circles. Each type of IBV was grouped in one circle and the representative strains were boxed. Phylogenetic trees were constructed with the neighbor-joining method using MEGA version 6.06. The bootstrap values were determined from 1000 replicates of the original data. The branch number represents the percentage of times that the branch appeared in the tree. Bootstrap values greater than 50% are shown. The *p*-distance is indicated by the bar at the bottom of the figure.

**Figure 2 viruses-11-00898-f002:**
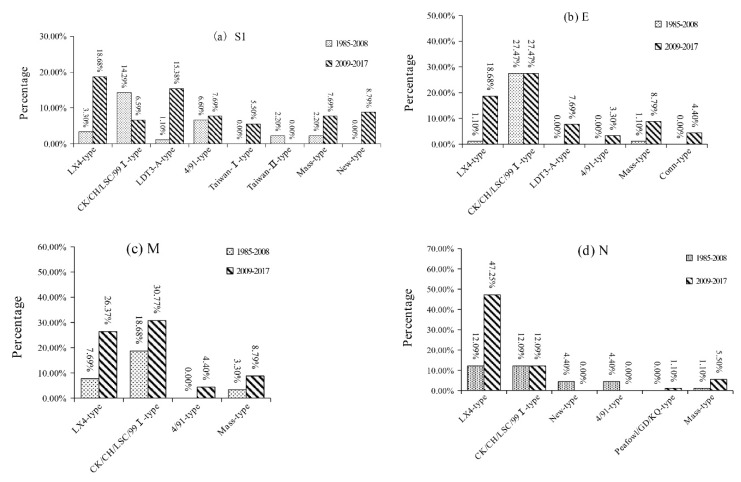
The percentages of different genotypes based on gene S1 (**a**), E (**b**), M (**c**), and N (**d**) of IBV strains isolated in different years.

**Figure 3 viruses-11-00898-f003:**
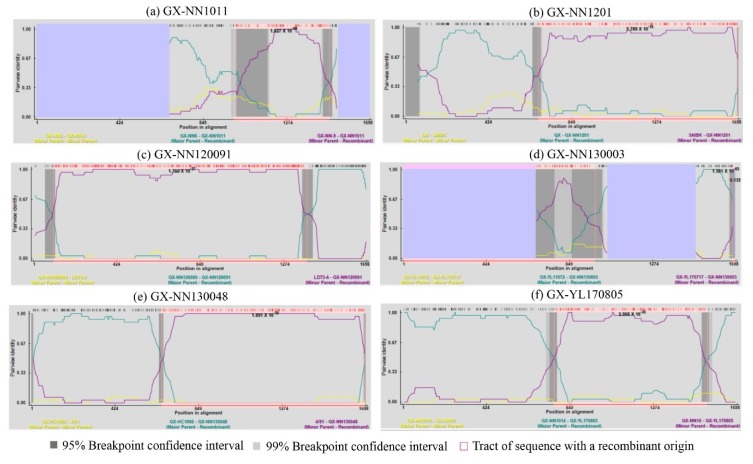
Recombination Detection Program (RDP) screenshots displaying the possible recombination events on the isolates (**a**) GX-NN1011, (**b**) GX-NN1201, (**c**) GX-NN120091, (**d**) GX-NN130003, (**e**) GX-NN130048, and (**f**) GX-YL170805. The highly similarity (>0.95) regions with putative parental fragments was show in dark gray area, and pink box indicated the recombinant origin with putative parental strains.

**Figure 4 viruses-11-00898-f004:**
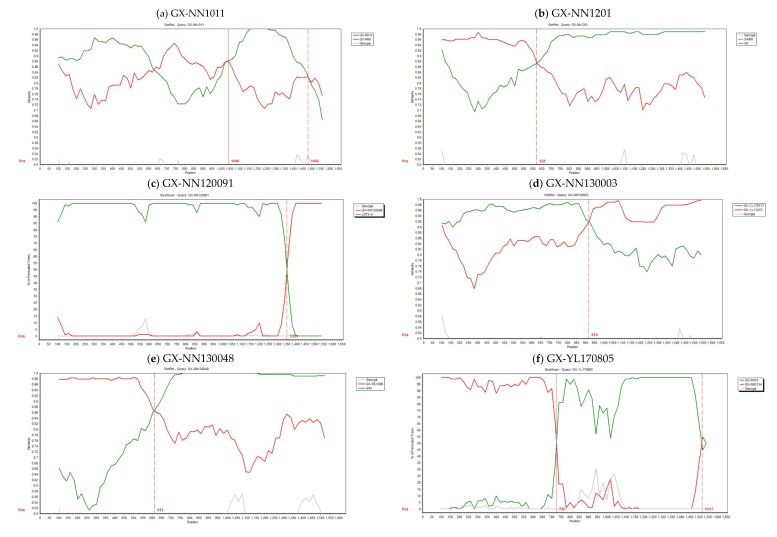
Similarity plots of the S1 structural gene region of GX-NN1011, GX-NN1201, GX-NN120091, GX-NN130003, GX-NN130048, and GX-YL170805 isolates created with SimPlot version 3.5.1. A 200-bp window with a 20-bp step was used. (**a**) GX-NN1011 isolate sequence used as the query, and GX-NN6 (red) and GX-NN-9 (green) used as putative parental strains. (**b**) GX-NN1201 isolate sequence used as the query, and QX (red) and SAIBK (green) used as putative parental strains. (**c**) GX-NN120091 isolate sequence used as the query, and GX-NN120089 (red) and LDT3-A (green) used as putative parental strains. (**d**) GX-NN130003 isolate sequence used as the query, and GX-YL11072 (red) and GX-YL170717 (green) used as putative parental strains. (**e**) GX-NN130048 isolate sequence used as the query, and GX-HC1006 (red) and 4/91 (green) used as putative parental strains. (**f**) GX-YL170805 isolate sequence used as the query, and GX-NN1014 (red) and GX-NN10 (green) used as putative parental strains.

**Figure 5 viruses-11-00898-f005:**
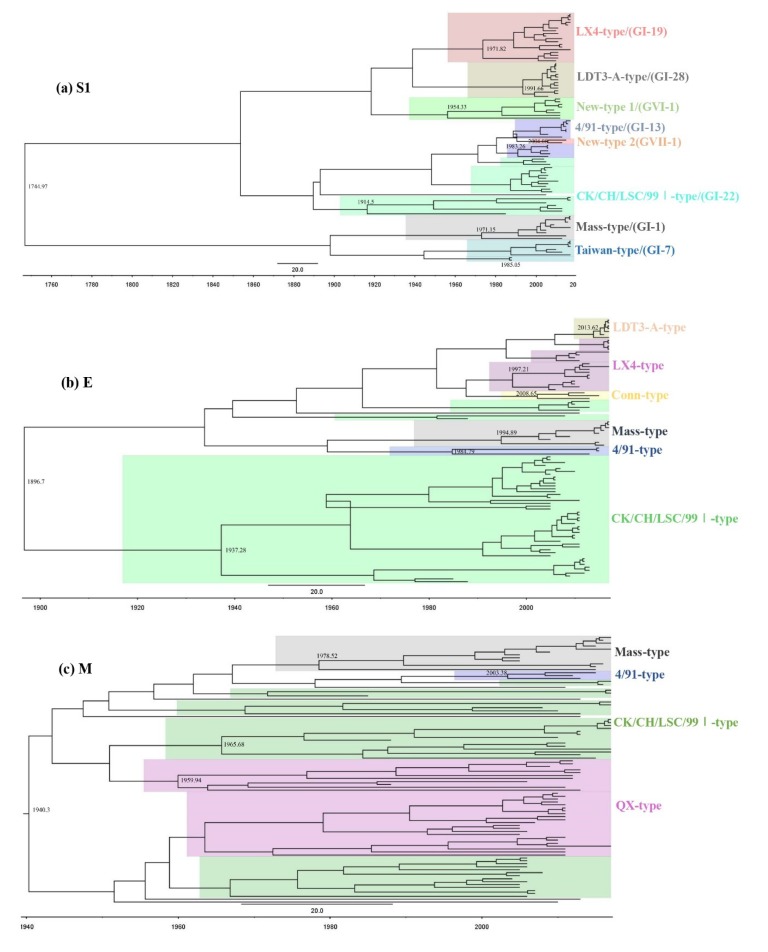
Time-resolved phylogenetic tree of the (**a**) S1, (**b**) E, (**c**) M, and (**d**) N genes of IBV using the Bayesian MCMC method (see Materials and Methods for further details). The MCC tree was constructed with 10% burn-in by Tree Annotator version 1.8.2 implemented in the BEAST software package. A change in branch color indicates a time change during the genotype’s evolutionary history. Numbers beside the branches are branching time. The scale bar represents the unit of time (year).

**Figure 6 viruses-11-00898-f006:**
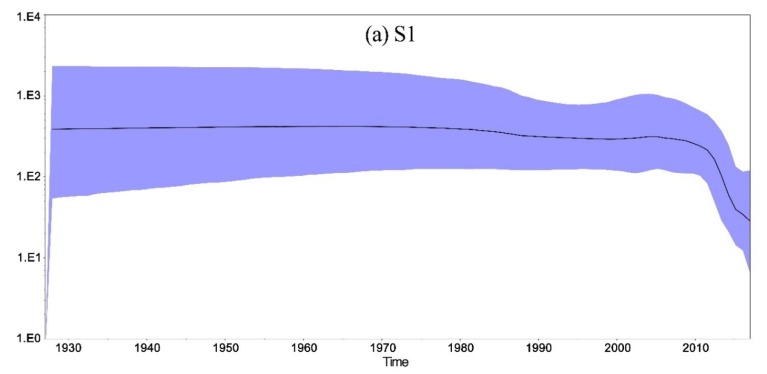
Bayesian skyline plot of (**a**) S1, (**b**) E, (**c**) M, and (**d**) N genes in IBV. The *Y*-axis represents the effective population size (log10 scale) and *X*-axis represents generation time (calendar year). The solid black line represents the mean value over time. The 95% HPD intervals are shown in blue region.

**Table 1 viruses-11-00898-t001:** Sixty-four IBV strains isolated during 2009–2017 in China.

IBV Isolates	Years of Isolation	Days of Age	Locations ^a^	GenBank Accession Numbers			
				S1	N	M	E
GX-NN09032	2009	N/A	Nanning	JX292013	JX567013	JX567012	KJ872796
GX-NN09093	2009	25	Nanning	JX292011	JX273225	JX273198	KJ940533
GX-GL	2010	N/A	Guilin	JX292008	JX273221	JX014370	KJ872786
GX-HC1006	2010	N/A	Hechi	JX292009	JX273229	JX273194	KJ940510
GX-NN1009	2010	N/A	Nanning	JX292000	JX273213	JX014366	KJ872793
GX-NN1011	2010	N/A	Nanning	JX292001	JX273216	JX014371	KJ940528
GX-NN1012	2010	N/A	Nanning	JX292002	JX273217	JX014372	KJ940529
GX-NN1013	2010	N/A	Nanning	JX292003	JX273218	JX014373	KJ940530
GX-NN1014	2010	N/A	Nanning	JX292004	JX273215	JX014367	KJ940531
GX-NN1019	2010	N/A	Nanning	JX292005	JX273219	JX014368	KJ872794
GX-GL11077	2011	17	Guilin	JX291992	JX273212	JX273186	KJ940507
GX-GL11078	2011	11	Guilin	JX291993	JX273209	JX273187	KJ940508
GX-GL11079	2011	N/A	Guilin	JX291994	JX273208	JX273188	KJ940509
GX-NN-4	2011	32	Nanning	JX291983	JX273199	JX273176	KJ940515
GX-NN-5	2011	77	Nanning	JX291984	JX273200	JX273177	KJ940517
GX-NN-6	2011	18	Nanning	JX291985	JX273207	JX273178	KJ940519
GX-NN-8	2011	27	Nanning	JX291986	JX273201	JX273179	KJ940520
GX-NN-9	2011	38	Nanning	JX291987	JX273202	JX273180	KJ940521
GX-NN-11	2011	45	Nanning	JX291988	JX273203	JX273182	KJ940523
GX-NN-13	2011	39	Nanning	JX291989	JX273204	JX273183	KJ940525
GX-NN-14	2011	34	Nanning	JX291990	JX273205	JX273184	KJ940526
GX-NN-15	2011	15	Nanning	JX291991	JX273206	JX273185	KJ940527
GX-NN11034	2011	12	Nanning	JX291999	JX273223	JX273191	KJ940535
GX-YL11072	2011	15	Yulin	JX291995	JX273210	JX273189	KJ940544
GX-YL11073	2011	14	Yulin	JX291996	JX273211	JX273190	KJ940545
GX-NN1201	2012	14	Nanning	JX436331	JX567014	JX567012	KJ940532
GX-NN120079	2012	35	Nanning	KJ999803	KF996273	KF996277	KM278995
GX-NN120084	2012	103	Nanning	KJ999804	KF996274	KF996278	KM278996
GX-NN120089	2012	43	Nanning	KJ999805	KF996275	KF996279	KM278997
GX-NN120091	2012	46	Nanning	KJ999806	KF996276	KF996280	KM278998
GX-NN130003	2013	22	Nanning	KJ999794	KF975404	KF996281	KM278999
GX-YL130025	2013	15	Yulin	KJ999795	KF975405	KF996283	KM279001
GX-NN130048	2013	22	Nanning	KJ999796	KF975406	KJ940503	KM279002
GX-YL13080630	2013	30	Yulin	KJ999797	KF975407	KF996286	KM279003
GX-YL130806200	2013	200	Yulin	KJ999798	KJ940498	KJ940501	KM279004
GX-NN130059	2013	15	Nanning	KJ999799	KF975408	KF996285	KM279005
GX-QZ131126	2013	80	Qinzhou	KJ999800	KJ940499	KJ940502	KM279006
GX-QZ130064	2013	35	Qinzhou	KJ999801	KJ940500	KJ940504	KM279007
GX-QZ130065	2013	37	Qinzhou	KT149876	KT188789	KT188788	KM279008
GX-NN130067	2013	52	Nanning	KJ999802	KJ940497	KJ940505	KM279009
GX-NN130021	2013	21	Nanning	KP085589	KF834569	KF996282	KM279000
GX-NN150019	2015	20	Nanning	MK887049	MK887118	MK887095	MK887072
GX-QZ150024	2015	13	Qinzhou	MK887057	MK887126	MK887103	MK887080
GX-YL150028	2015	20	Yulin	MK887060	MK887129	MK887106	MK887083
GX-LZ150619	2015	19	Liuzhou	MK887046	MK887115	MK887092	MK887069
GX-YL150727	2015	73	Yulin	MK887061	MK887130	MK887107	MK887084
GX-LZ160322	2016	10	Liuzhou	MK887047	MK887116	MK887093	MK887070
GX-YL161022	2016	15	Yulin	MK887063	MK887132	MK887109	MK887086
GX-YL161015	2016	100	Yulin	MK887062	MK887131	MK887108	MK887085
HN-170502	2017	90	Hainan	MK887045	MK887114	MK887091	MK887068
GX-LZ170609	2017	43	Liuzhou	MK887048	MK887117	MK887094	MK887071
GX-YL170717	2017	105	Yulin	MK887064	MK887133	MK887110	MK887087
GX-YL170718	2017	24	Yulin	MK887065	MK887134	MK887111	MK887088
GX-QZ170728	2017	194	Qinzhou	MK887058	MK887127	MK887104	MK887081
GX-YL170805	2017	24	Yulin	MK887066	MK887135	MK887112	MK887089
GX-YL170808	2017	140	Yulin	MK887067	MK887136	MK887113	MK887090
GX-NN170502	2017	118	Nanning	MK887050	MK887119	MK887096	MK887073
GX-NN170829	2017	27	Nanning	MK887052	MK887121	MK887098	MK887075
GX-NN170825	2017	11	Nanning	MK887051	MK887120	MK887097	MK887074
GX-QZ171023	2017	9	Qinzhou	MK887059	MK887128	MK887105	MK887082
GX-NN170901	2017	20	Nanning	MK887053	MK887122	MK887099	MK887076
GX-NN171108	2017	52	Nanning	MK887054	MK887123	MK887100	MK887077
GX-NN171123	2017	60	Nanning	MK887055	MK887124	MK887101	MK887078
GX-NN171125	2017	42	Nanning	MK887056	MK887125	MK887102	MK887079

N/A data not available; ^a^ Area where the viruses were isolated.

**Table 2 viruses-11-00898-t002:** Genetic recombination events of IBV isolates detected by RDP 4 software.

Potential Recombinant	Breakpoints		Major Parent (Similarity)	Minor Parent (Similarity)	Detection Method ^a^
	Beginning	Ending			
GX-NN1011	1040	1482	GX-NN6 (LX4-type) (91.1%)	GX-NN-9 (LX4-type) (95.1%)	RDP, GENECONV, MaxChi, Chimera, SiScan, 3Seq
GX-NN1201	621	1612	QX (LX4-type) (96%)	SAIBK (CK/CH/LSC/99I-type) (98.5%)	RDP, GENECONV, Bootscan, MaxChi, Chimera, SiScan, 3Seq
GX-NN120091	100	1351	GX-NN120089 (New-type 1) (99.2%)	LDT3-A (99.1%)	RDP, GENECONV, Bootscan, MaxChi, SiScan, 3Seq
GX-NN130003	1	916	GX-YL11072 (LDT3-A-type) (96.9%)	GX-YL170717 (CK/CH/LSC/99I-type) (95.8%)	MaxChi, Chimera, SiScan, 3Seq
GX-NN130048	611	1616	GX-HC1006 (LX4-type) (98.1%)	4/91 (99.6%)	RDP, GENECONV, Bootscan, MaxChi, SiScan, 3Seq
GX-YL170805	725	1517	GX-NN1014 (Taiwan-I-type) (97.8%)	GX-NN10 (4/91-type) (98.5%)	RDP, GENECONV, Bootscan, MaxChi, Chimera, SiScan, 3Seq

^a^ Only transferred gene fragments where *p* ≤ 1 × 10^−6^ are included in the table. The major parent is the sequence most closely related to that surrounding the transferred fragment. The minor parent is most closely related to the transferred fragment in the recombinant.

**Table 3 viruses-11-00898-t003:** Selection profile of S1, E, M, and N proteins of Guangxi IBV isolates.

Model	S1 Protein			E Protein			M Protein			N Protein		
	SLAC	FEL	IFEL	SLAC	FEL	IFEL	SLAC	FEL	IFEL	SLAC	FEL	IFEL
Mean dN/dS	0.283			0.266			0.146			0.147		
Numbers of positive selection site	13	13	15	1	8	7	5	5	6	6	11	9
Numbers of neutral selection site	366	277	287	85	65	69	136	119	127	246	196	209
Numbers of purifying selection site	178	267	255	33	46	43	85	102	93	157	202	191
Positively selected sites (aa)	19, 39, 106, 119, 155, 503			119			12, 16, 17, 47, 151			7, 10, 235, 342, 345		

* SLAC (single-likelihood ancestor counting), FEL (fixed effects likelihood), and IFEL (internal fixed effects likelihood).

**Table 4 viruses-11-00898-t004:** Prediction results of IBVs by NetNGlyc 1.0 Server.

Protein	Position	Sequence	Potential	Agreement	N-Glyc Result	N-linked Gylcosylation Sites/Strains	Percentage/%
S1	239/240 ^a^ 427/428 ^b^	NFSD NITL	0.5840–0.5851 0.5598–0.6254	(7/9) (7/9)	+ +	14 (3/91)	3.30
						15 (3/91)	3.30
						16 (1/91)	1.10
						17 (3/91)	3.30
						18 (18/91)	19.78
						19 (29/91)	31.87
						20 (15/91)	16.48
						21 (17/91)	18.68
						22 (2/91)	2.20
E	11/12 5/6 ^c^	NGSF NKSL	0.5228–0.5336 0.6297–0.7593	(6/9) (9/9)	+ ++	1 (2/91)	1.10
						2 (49/91)	53.85
						3 (40/91)	43.96
M	3/4/6	NCTL	0.6007–0.7913	(7/9)/ (9/9)	+ ++	1 (7/91)	7.69
						2 (84/91)	92.31
N	32 ^d^	NASW	0.5564–0.7239	(6/9)/ (9/9)	+ ++	0 (1/91)	1.10
						1(38/91)	41.96
						2 (51/91)	56.04
						3 (1/91)	1.10

^a^ except CK/CH/LSC/99I-type strains; ^b^ except Taiwan-type and New-type strains; ^c^ except CK/CH/LSC/99I-type and Conn-type strains; ^d^ except GX-NN120084 strain.

**Table 5 viruses-11-00898-t005:** Estimate of evolutionary rates (nucleotide substitutions/site/year) of IBV strains.

Structural Genes	Mutation/Evolutionary Rates (Substitutions/Site/Year)	
	Mean	95 % HPD
S1	4.6 × 10^−3^	3.7 × 10^−3^–5.5 × 10^−3^
E	4.3 × 10^−3^	2.8 × 10^−3^–5.8 × 10^−3^
M	3.9 × 10^−3^	1.2 × 10^−3^–4.6 × 10^−3^
N	3.7 × 10^−3^	2.7 × 10^−3^–4.7 × 10^−3^

HPD, highest probability density.

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
