# Peer review of "Genetic Analysis of Avian Coronavirus Infectious Bronchitis Virus in Yellow Chickens in Southern China over the Past Decade: Revealing the Changes of Genetic Diversity, Dominant Genotypes, and Selection Pressure"

_viruses, 2019, doi:10.3390/v11100898_

Round 1
Reviewer 1 Report
The manuscript presents a very comprehensive picture of the variation of the genes encoding structural proteins of IBV strains recently (2009-2017) infecting chickens in China. IBV is a wonderful example of the variations of a virus, with a large RNA genome, that can occur in highly dense populations. In addition to mutation rates, the authors have presented valuable trends. This study is particularly unique in that it includes sequences encoding 4 structural genes. Although not mentioned, their study really would imply that vaccines should be nonreplicating, if not designed to eliminate recombination. The latter should be incorporated into their discussion.
There are a number of grammatical errors that need to be addressed and will make this large manuscript easier to read. Most are indicated below.
The font size frequently changes throughout the manuscript.
Introduction
Line 51, could not explain
This is not the appropriate wording. Maybe “causes confusion”??
Line 56 “a total of seven genotypes”
Line 64 “then becoming the”
Line 67 “based on the S1”
Line 67 “differences in breeding variabilities and feeding patterns
Line 68 “at different times and in different regions”
Line 71 “rates”
Line 73 “is not driven”
Line 83 “closed environment”
Line 84 “with different chicken breeds and differing”
Line 88 “has been a continuing problem”
Line 90 “IBVs from 1985-2008”
Materials and Methods
Line 112 “were the same as previously described”
Line 120 “with the exception of forty-two strains”
Line 122 “were also analyzed together”
Line 127 “scientificalness??” What is this word?
Lines 127-129; This sentence needs to be rewritten.
Results
Line 197 “filled black triangles”
Line 198 “filled red circles.” “IBV was”
Line 211 “was the predominant”
Line 280 “In addition” What’s more is not an acceptable phrase.
Line 281 “values of larger than”
Discussion
Line 331 “is the largest region”
Line 334 “of the entire S1, E, M and N genes analyzed.”
Line 336 “from 1985-2008”
Line 342 “have resulted in great challenges controlling”
Line 344 “a total of “
Line 381 “and LZ4 (QX) causing severe economic losses have been”
Line 385 “Therefore, monitoring of wild birds”
Line 386 “also need to be followed.”
Line 387 “and has in recent years spread to many chicken flocks”
Line 392 “there was a four aa (QKEP) insertion located in the HVR III and one aa(S) insertion in all the New-type 1 isolates.”
Line 397 “isolates were shown to undergo recombination in our study”
Line 399-400 “this first report of a recombination event needs to be further investigated.”
Line 445 “Despite that the four”
Lines 466-472. I do not understand how the dates of origin were calculated. Since these dates go back several hundred years such an assertion should be well explained providing the information is in this study.
In addition, this paragraph is pretty rough and should be rewritten.
Line 479 “Therefore, the 4/91-like vaccine”
Line 483 “in other recent studies”
Line 493 “will become more and more intense.”
Line 498 “there has been a remarkable”
Line 502 “suggest a complexity”
Line 506 “also suggests the importance”
Line 510 popular is the wrong word. Maybe prevalent.
Line 519 “that the IBV strains in southern China have”
Conclusions: New strains should be resistant to recombination and mutation.

Author Response
Dear reviewer:
Responses to the comments of reviewer 1
Thank you very much for your favorable, precious comments and suggestions. We have carefully considered the comments and revised the manuscript accordingly. The English language and style of our manuscript have been checked.
Comments and Suggestions for Authors
- The manuscript presents a very comprehensive picture of the variation of the genes encoding structural proteins of IBV strains recently (2009-2017) infecting chickens in China. IBV is a wonderful example of the variations of a virus, with a large RNA genome, that can occur in highly dense populations. In addition to mutation rates, the authors have presented valuable trends. This study is particularly unique in that it includes sequences encoding 4 structural genes. Although not mentioned, their study really would imply that vaccines should be nonreplicating, if not designed to eliminate recombination. The latter should be incorporated into their discussion.
> Thank you very much for your favorable, precious comments. We have incorporated your suggestion into the discussion (line 441-443).
- There are a number of grammatical errors that need to be addressed and will make this large manuscript easier to read. Most are indicated below.
> The English language and style of our manuscript have been checked carefully and corrected.
- The font size frequently changes throughout the manuscript.
> We checked the manuscript carefully to make sure that the font size was consistent throughout the manuscript.
Introduction
- Line 51, “could not explain” This is not the appropriate wording. Maybe “causes confusion”??
> Thanks for the suggestion. But we prefer to keep the wording“could not explain”, because such wording could express the real meaning. In addition, we corrected “causes the misunderstanding” into “is misleading” (line 53).
- Line 56 “a total of seven genotypes”
> We corrected “total seven genotypes” into “a total of seven genotypes” as recommended (line 57).
- Line 64 “then becoming the”
> We corrected “then become the” into “then becoming the” as recommended (line 65).
- Line 67 “based on the S1”
> We corrected “based on S1” into “based on the S1” as recommended (line 68).
- Line 67 “differences in breeding variabilities and feeding patterns”
> We corrected “difference in breeding varieties and feeding patterns” into “differences in breeding variabilities and feeding patterns” as recommended (line 68).
- Line 68 “at different times and in different regions”
> > We corrected “in different times and regions” into “at different times and in different regions” as recommended (line 69-70).
- Line 71 “rates”
> We corrected “a high mutation and recombination rate” into “high mutation and recombination rates” as recommended (line 72).
- Line 73 “is not driven”
> We corrected “was not driven” into “is not driven” as recommended (line 74).
- Line 83 “closed environment”
> We corrected “closed” into “closed environment” as recommended (line 84).
- Line 84 “with different chicken breeds and differing”
> We corrected “different chicken breeds with different” into “with different chicken breeds and differing” as recommended (line 85).
- Line 88 “has been a continuing problem”
> We corrected “has been continually a popular problem” into “has been a continuing problem” as recommended (line 90).
- Line 90 “IBVs from 1985-2008”
> We corrected “IBVs during 1985-2008” into “IBVs from 1985 to 2008” as recommended (line 91).
Materials and Methods
- Line 112 “were the same as previously described”
> We corrected “were same as previous description” into “were the same as previously described” as recommended (line 113-114).
- Line 120 “with the exception of forty-two strains”
> We corrected “excepted forty-two strains” into “with the exception of forty-two strains” as recommended (line 121).
- Line 122 “were also analyzed together”
> We corrected “were also used to analyze together” into “were also analyzed together” as recommended (line 123).
- Line 127 “scientificalness??” What is this word?
> We corrected “scientificalness” into “scientificity” (line 128).
- Lines 127-129; This sentence needs to be rewritten.
> We corrected “To ensure the scientificalness and reliability of the results, the reference sequences of S1, E, M and N genes of each strain were extracted from the individual reference strains with complete genome sequences as much as possible.” into “To ensure the scientificity and reliability of the results, we extracted the S1, E, M and N genes from the complete genome sequences of reference strains.” as recommended (line 128-130).
Results
- Line 197 “filled black triangles”
> We corrected “filled black triangle” into “filled black triangles” as recommended (line 200).
- Line 198 “filled red circles.” “IBV was”
> We corrected “filled red circle” and “IBVs was” into “filled red circles” and “IBV was”, respectively as recommended (line 201).
- Line 211 “was the predominant”
> We corrected “were the predominant genotypes” into “was the predominant genotype” as recommended (line 214).
- Line 280 “In addition” What’s more is not an acceptable phrase.
> We corrected “What’s more” into “In addition” as recommended (line 284).
- Line 281 “values of larger than”
> We corrected “values of bigger than” into “values of larger than” as recommended (line 285).
Discussion
- Line 331 “is the largest region”
> We corrected “is the biggest region” into “is the largest region” as recommended (line 341).
- Line 334 “of the entire S1, E, M and N genes analyzed.”
> We corrected “of the entire S1, E, M and N genes.” into “of the entire S1, E, M and N genes analyzed.” as recommended (line 344).
- Line 336 “from 1985-2008”
> We corrected “during 1985-2008” into “from 1985 to 2008” as recommended (line 346).
- Line 342 “have resulted in great challenges controlling”
> We corrected “brought great challenges for the control of IB through the vaccination measure.” into “have resulted in great challenges for the controlling IB through vaccination.” as recommended (line 352-353).
- Line 344 “a total of”
> We corrected “Total” into “A total of” as recommended (line 355).
- Line 381 “and LX4 (QX) causing severe economic losses have been”
> We corrected “and LX4 (QX) strain have been reported and caused severe economic losses” into “and LX4 (QX) causing severe economic losses have been reported” as recommended (line 391-392).
- Line 385 “Therefore, monitoring of wild birds”
> We corrected “So, monitoring of wild birds” into “Therefore, monitoring of wild birds” as recommended (line 395).
- Line 386 “also need to be followed.”
> We corrected “also needed to detect.” into “also need to be followed.” as recommended (line 397).
- Line 387 “and has in recent years spread to many chicken flocks”
> We corrected “and then spread to many chicken flocks in recent years” into “and has in recent years spread to many chicken flocks” as recommended (line 398-399).
- Line 392 “there was a four aa (QKEP) insertion located in the HVR III and one aa (S) insertion in all the New-type 1 isolates.”
> We corrected “there were four aa (QKEP) located in the HVR Ⅲ and one aa (S) insertions in all the New-type 1 isolates.” into “there was a four aa (QKEP) insertion located in the HVR III and one aa (S) insertion in all the New-type 1 isolates.” as recommended (line 403-404).
- Line 397 “isolates were shown to undergo recombination in our study”
> We corrected “isolates underwent recombination events in our study” into “isolates were shown to undergo recombination in our study” as recommended (line 409).
- Line 399-400 “this first report of a recombination event needs to be further investigated.”
> We corrected “the first report on recombination event of New-type 1 isolate. And we need to investigate this event further.” into “this first report of a recombination event needs to be further investigated.” as recommended (line 411-412).
- Line 445 “Despite that the four”
> We corrected “Despite the four” into “Despite that the four” as recommended (line 460).
- Lines 466-472. I do not understand how the dates of origin were calculated. Since these dates go back several hundred years such an assertion should be well explained providing the information is in this study.
In addition, this paragraph is pretty rough and should be rewritten.
> The dates of origin were calculated by the Bayesian Markov chain Monte Carlo (MCMC) method. The detail method was descripted in the “Materials and Methods” section (line 157-172) and cited from the references [30], [39] and [40].
We rewritten the paragraph and discussed about “Since these dates go back several hundred years such an assertion should be well explained providing the information is in this study” as recommended (line 479-489).
- Line 479 “Therefore, the 4/91-like vaccine”
> We corrected “Therefore, 4/91-like vaccine” into “Therefore, the 4/91-like vaccine” as recommended (line 505).
- Line 483 “in other recent studies”
> We corrected “in other studies recently” into “in other recent studies” as recommended (line 508).
- Line 493 “will become more and more intense.”
> We corrected “will become more and more heavier.” into “will become more and more intense.” as recommended (line 520).
- Line 498 “there has been a remarkable”
> We corrected “there is a remarkable” into “there has been a remarkable” as recommended (line 524).
- Line 502 “suggest a complexity”
> We corrected “suggest the complexity” into “suggest a complexity” as recommended (line 528).
- Line 506 “also suggests the importance”
> We corrected “also means that the importance” into “also suggests the importance” as recommended (line 532).
- Line 510 popular is the wrong word. Maybe prevalent.
> We corrected “popular” into “prevalent” as recommended (line 536).
- Line 519 “that the IBV strains in southern China have”
> We corrected “that the IBVs in southern China” into “that the IBV strains in southern China” as recommended (line 545).
- Conclusions: New strains should be resistant to recombination and mutation.
> We corrected it as recommended (line 548-549).
Reviewer 2 Report
The authors extensively analyzed genome sequences of IBV using multiple softwares to track the genetic evolution. The study design and methods are scientifically sound and the results provide significant information on the viral genetic evolution as well as for vaccine development. I have only a few comments on the data presentation and format as follows;
- Size of legend texts in Figures are too small to recognize. This need to be edited.
- Figure 1: The authors need to provide information on how they define the each groups. Please comment on the criteria of nodal value to define groups.
- Figure 3, Table 3&4: I'd like to suggest to present schematic diagrams of each ORF including the location of positive selection sites, putative glycosylation sites, and recombination break points. In addition, indication of predicted domain structures in each ORF will gives better insight on the evolutionary pathways in functional domain levels.
Author Response
Dear reviewer:
Responses to the comments of reviewer 2
Thank you for your favorable, precious comments and suggestions. We have carefully considered the comments and revised the manuscript accordingly. The English language and style of our manuscript have been checked.
Comments and Suggestions for Authors
- Size of legend texts in Figures are too small to recognize. This need to be edited.
> We edited the size of legend texts in Figures and re-uploaded them.
- Figure 1: The authors need to provide information on how they define the each groups. Please comment on the criteria of nodal value to define groups.
> We defined the each group according to the reference [18]. (Valastro, V.; Holmes, E.C.; Britton, P.; Fusaro, A.; Jackwood, M.W.; Cattoli, G.; Monne, I. S1 gene-based phylogeny of infectious bronchitis virus: An attempt to harmonize virus classification. Infect Genet Evol. 2016, 39, 349-364.) Different genotypes should have an average distance per site >10% (0.1). Different sub-genotypes should have an average distance per site between 3 (0.03) and 10% (0.1). Because calculating the p-distance between different sequences are common in IBV articles, we added p-distance values between the groups using the Maximum Composite Likeli-hood method from the references [18] in MEGA 6. We added Supplementary Table S2-S5 to provide the information on how we define the each groups of S1, E, M and N gene (line 211, line 550-554).
- Figure 3, Table 3&4: I'd like to suggest to present schematic diagrams of each ORF including the location of positive selection sites, putative glycosylation sites, and recombination break points. In addition, indication of predicted domain structures in each ORF will gives better insight on the evolutionary pathways in functional domain levels.
> Thanks for the good suggestion. But there were so many IBV strains, if we present schematic diagrams of each ORF including the location of positive selection sites, putative glycosylation sites, and recombination break points, the recombination break points will not be as intuitive as Figure 3. Therefore, we hope to keep the Figure 3 to present recombination break points, but we tagged the recombination break points in Supplementary Figure S2. Similarly, we hope to keep the Table 3&4 and present schematic diagrams of each ORF including the location of positive selection sites and putative glycosylation sites respectively, but we added schematic diagrams of each ORF including the location of positive selection sites and putative glycosylation sites in Supplementary Figure S4-S7 (line 562-569).
“In addition, indication of predicted domain structures in each ORF will gives better insight on the evolutionary pathways in functional domain levels.” This is really good suggestion. But only one strain can be analyzed at a time. We had 64 IBV isolates and there will be a lot of graphics in the manuscript. Hence, we plan to exclude this analysis from the manuscript, but this will be our work in the near future.